# Article
# Life Cycle Assessment in Higher Education: Design and Implementation of a Teaching Sequence Activity

Alberto Navajas [1,2], Itsaso Echarri [1], Luis M. Gandía [1,2], Jorge Pozuelo [3] and Esther Cascarosa [3,*]

1 Department of Science, Public University of Navarre (UPNA), Arrosadía Campus s/n, 31006 Pamplona, Spain; alberto.navajas@unavarra.es (A.N.); itsaso.echerri@unavarra.es (I.E.); lgandia@unavarra.es (L.M.G.)
2 Institute for Advanced Materials and Mathematics (InaMat2), Universidad Pública de Navarra (UPNA), Campus de Arrosadia, 31006 Pamplona-Iruña, Spain
3 Department of Specific Didactics, University of Zaragoza, c/Pedro Cerbuna 12, 50009 Zaragoza, Spain; jpozuelo@unizar.es
* Correspondence: ecascano@unizar.es

**Abstract:** The latest studies show that to achieve the Sustainable Development Goals on education, there must be a focus on adequately training higher education students. In this work, we present a study about the Life Cycle Analysis of knowledge of products and processes of engineering students. This aspect is very relevant in engineering education since it has direct implications on sustainability. The first step was to identify what the learning problems were, and taking them into account, a specific teaching sequence was designed and implemented over three academic years. Two activities, on an increasing level of complexity, of the application of Life Cycle Assessment are shown in this paper. The first one is the Life Cycle Analysis comparison between two steel and polypropylene pieces. The second one is the Life Cycle Analysis comparison between three different ends of life of a polypropylene piece: mechanical recycling, incineration, and landfill. Data on the evolution of students' marks while solving a "one step more difficult project" throughout these courses have been collected. The results show a generalized learning by the students about Life Cycle Analysis.

**Keywords:** life cycle assessment teaching; life cycle inventory; engineering education; education teaching activities





## 1. Introduction

From the field of study of this work, chemical engineering sustainability is closely related to the life cycle of products and processes. Knowing how to optimally evaluate (and calculate) the life cycle of something will avoid replacing parts or modifying processes, which would entail a high cost in materials with what this means for the environment and natural resources in general. This work arises from the need to evaluate whether it is possible to teach Life Cycle Assessment (LCA) in a way that allows students to have a deeper understanding of the process. The need for this study and the results obtained are substantiated below.

According to some authors [1–3], the design of education at a global level is not addressing the SDGs. However, these same authors put the focus of hope on higher education. Owen and Chankseliani and McCowan [4,5] explained that higher education plays a key role as a means to achieve goal number 4 on education. And one way to achieve it is by making students more aware of the need for sustainable processes, especially those developed in industry, which is the context of students who are training in engineering. Therefore, it is relevant to design university education that promotes quality (thorough SDG number 4).

Chemical engineering is one of the most important branches of industry since it provides a large number of products to society that are vital for its development and, therefore,



it is a degree closely related to sustainability. Due to the climate and resources crisis, chemical engineering must find solutions to provide society with sustainable products, that is, with the same quality but with less consumption of resources and emissions. To achieve this goal, chemical engineering must integrate the concept of the life cycle into systems and products and try to bring linear industrial products closer to cycles that generate the least amount of pollutants and wastes, and use these wastes as resources. For this reason, one scientific tool that has become essential for chemical engineers is the LCA that quantifies the environmental impacts of processes and products along their life cycles. By using this tool, it can be determined which stage of the life cycle of a process, or product, has the highest environmental impact and provide solutions to reduce it. As a result, and with the aim of training qualified professionals to solve real problems, during the last few years, LCA has become a mandatory subject in several chemical engineering university grade curriculums [6–9].

LCA has become so important that industries, administration, educational, and scientific communities usually include it in their projects and products to quantify the environmental impacts as a regular procedure [10]. Most of the LCA studies are performed with commercial software that are very useful when carrying out LCAs since they contain large databases and automatically transform the elementary inputs and outputs into Environmental Impact Indicators (EIIs). However, they also have the problem of performing all the LCA operations, omitting the fundamental steps such as the transformation of data to environmental indicators or the use of concepts as essential as the characterization factors or functional unit. Therefore, many times, LCA software (GaBi® Educational version v2022.2) users do not really know the steps that it performs or the basic concepts behind their results [11,12]. This often leads to a lack of in-depth understanding of the results obtained, which is crucial for the training of higher education students.

For these reasons, in this work, we have analyzed how to introduce a teaching sequence where the calculations of the LCA process are detailed, so that, although engineering students use calculation programs in their future work, they have a robust training in the calculation underlying those programs.

Some pedagogical LCA experiences on chemical and material engineering courses are described in the literature [13–19]. However, there is a lack of teaching material in the scientific literature or textbooks that provide examples of the basic LCA calculations made in spreadsheets. One possible reason for this, as pointed out by Burnley et al. [20], is that LCA is taught in many higher education institutions, but most of them do not publish the didactic material they use in their lessons. One of the few studies that provided the basic LCA calculations was Cosme et al. [21], although they were only for the related inventory to functional unit and not to calculate the EIIs from elementary flows. Furthermore, the calculations were made for isolated examples and not for an example that followed the idea of "red thread" as previously cited. There are also several LCA textbooks where LCA theory was explained [22–28] but these textbooks lack LCA calculations from the inventory to the EIIs. Maybe the only one that makes the calculations and provides some material to the teachers was the Wimmer and Züst [27] textbook where the ecodesign of a water kettle was performed via LCA.

The objective of this article is two-fold: on the one hand, based on the necessity described, two examples of the application of LCA have been designed in a very detailed way. On the other hand, to analyze the impact of the application of these activities' sequence into the student´s marks over three years of teaching.

The researchers (authors of this article) have been teachers of the course for ten years. Therefore, the historical evolution of this course during this time can be described in order to introduce the necessity of this work. Since 13/14, the subject "Ecodesign" has been structured in a theoretical part in one half and the other half practical. The practical part has consisted of practices with the LCA GaBi® Education v2022.2 software in which students develop their own ecodesign projects using the LCA tool. During these years, the aforementioned phenomenon of a lack of basic knowledge of fundamental concepts of LCA

has been observed in students, which also resulted in low marks. The basic knowledge of the students was limited to reproducing the calculations with the LCA tools; however, when the context of the problem changed, they did not know how to solve it. These results were indicative of a lack of deep knowledge of the fundamental concepts that prevented the final step of the application of said concepts (generalization of learning).

As far as the authors know, this is the first time LCA teaching tools and spreadsheets with calculations are provided to assess engineering and science educational professionals regarding the correct understanding of the theoretical LCA concepts of their students.

## 2. Materials and Methods

### 2.1. Sample

The participants in this study were 34 students of "Ecodesign", which is an optional course in the fourth academic year of the "Industrial Mechanical Design Engineer" university degree. This is a representative sample of the total number of students in the last few years. All of them were informed about the anonymous treatment of their marks in the subject, in order to be able to use them within this work.

### 2.2. Procedure

To achieve the two objectives set out in this study, on the one hand, the researchers proposed designing a sequence (composed of two steps) to facilitate the understanding of the calculations that were previously carried out using LCA tools. This design has taken care of even the smallest calculation, with the aim of cushioning the lack of knowledge in the steps. And on the other hand, to evaluate the effectiveness of this procedure, evaluation data have been collected at two moments in the process.

Since the 20/21 academic course, the methodology and examples of ecodesign were modified in order to facilitate deep learning about LCA for the students. As a final part of this development, the sequences of teaching relating to LCA that are presented in this article have been designed and implemented during three consecutive academic years (20/21 to 22/23) following the "red tread" idea and explain in detail the different LCA steps.

During the 60 h of on-site lessons, the teaching sequence designed is introduced within the 30 first part hours. Once this first part ends, the students complete a midterm exam. In this written exam, students were asked about solving LCA cases. The second part of the lessons (30 h) are practical and in front of the computer, which has the LCA software (GaBi® Educational version v2022.2) installed, and at the end of this part, the students must develop a project from which they obtain the final mark. This final project consists of a generalized application of their learnings where the students have to compile the elementary flows related to the fundamental flows and the functional unit; to convert the elementary flows into Environmental Impact Indicators (EIIs); to show and interpret the LCA results. That is a summary of all the treatments along the course. These projects were explained to the group and the researcher took observational notes.

## 3. Results

### 3.1. Sequence of Teaching Activities

The first LCA example is a comparison between two steel and polypropylene pieces. The second LCA example is a comparison between three different ends of life of a polypropylene piece: mechanical recycling, incineration, and landfill. This second example increases the calculations' complexity and serves to explain the open-loop recycling of the steam and electricity generated on incineration and recycled PP granulate generated on a mechanical recycling process. The two LCA examples are explained step by step, with "paper, pen and calculator", and using Excel sheets provided in the Supplementary Materials of this article. The four steps that ISO 14040 and 14044 [29,30] marked as essential for an LCA will be carried out. These steps are avoided by the LCA software (GaBi® Educational version v2022.2) but are essential for the correct understanding of LCA concepts. To show where

the results come from and thus avoiding simplistic interpretations, special emphasis is placed on the calculations.

Therefore, this section is going to be divided into the two LCA examples and into the four steps that make up the LCA: 1. definition of the objective and scope of the study; 2. Life Cycle Inventory (LCI); 3. Life Cycle Impact Assessment (LCIA); and 4. interpretation of the results.

### 3.1.1. Steel and Polypropylene Pieces LCA Example
Scope and Objective of the LCA

ISO 14044 [30] stipulated that in this first step, the LCA goal, functional unit used, limits of the study, EIIs used, and data sources have to be described. Regarding the goal, this LCA is going to determine the environmental impacts of the two different pieces of steel and polypropylene (PP) used. Specifically, the students have to solve the next problem: "One company wants to substitute one steel machinery piece by one of polypropylene. Steel piece weight is 2.3 kg and its lifetime is 1.5 years. Polypropylene piece weight is 0.75 kg and its lifetime is 0.56 years. For a machine working time of 20 years, determine which piece has less environmental impacts by LCA".

The functional unit is the magnitude with which the reference flows of the system are calculated. Thus, the functional unit of this study is the use of the pieces (steel or PP) over 20 years. This study is carried out using a very simplified version of the databases provided by the educational version of GaBi Software v2022.2 and by the scientific literature. Regarding the inventories of the materials obtained in GaBi, the inputs and outputs that generate the greatest impact in each indicator have been compiled. The 16 EIIs recommended by the EU to carry out LCAs [31] and which are detailed in Table 1 are used. The first 12 EIIs take into account the impacts of the elementary outputs, and the 4 last impacts take into account the elementary inputs. In this study, only the manufacture and use stages of the life cycle of the pieces are going to be considered.

**Table 1.** Environmental impact indicators.

| Environmental Impact Indicator | Abbreviation | Unit |
| --- | --- | --- |
| Climate change | GWP | kg $CO_2$ eq. |
| Ozone depletion | ODP | kg CFC-11 eq. |
| Respiratory inorganics | RI | Disease incidences |
| Ionizing radiation—human health | IR | kBq U235 eq. |
| Photochemical ozone formation—human health | POF | kg NMVOC eq. |
| Acidification terrestrial and freshwater | AC | Mole of H+ eq. |
| Eutrophication terrestrial | EUT | Mole of N eq. |
| Eutrophication freshwater | EUF | kg P eq. |
| Eutrophication marine | EUM | kg N eq. |
| Cancer human health effects | HTC | CTUh |
| Non-cancer human health effects | HTNC | CTUh |
| Ecotoxicity freshwater | ECFW | CTUe |
| Land use | LU | Pt |
| Resource use. Mineral and metals | RDM | kg Sb eq. |
| Water scarcity | WU | $M^3$ world equiv. |
| Resource use. Energy carrier | RU | MJ |

Life Cycle Inventory (LCI)

ISO 14040 [29] stipulated "inventory analysis involves data collection and calculation procedures to quantify relevant inputs and outputs of a product system". This is the most complex stage of LCA. However, the use of software has made it much easier for LCA professionals since their databases provide extensive industrial processes with a high amount of reliable data. As has been said in the introduction, the most counterproductive part of this software is that it avoids calculating the adequacy of these data to the fundamental flow units (fluxes that link processes inside the technosphere) for each functional unit. By

this way, the LCA software (GaBi® Educational version v2022.2) eliminates a degree of knowledge about the system. Therefore, it is necessary to carry out the necessary operations to go from the data provided by the databases to the final compilation of the elementary inputs and outputs (from the ecosphere to the technosphere and vice versa) for a good understanding of the calculations made to obtain the Life Cycle Inventory.

That is the reason why, in this study, with a theoretical LCA teaching character, these calculations are carried out with only the help of a calculator and Excel files. Firstly, the number of pieces of each material are calculated, as follows: 13.33 steel pieces (20 years/1.5 years each steel piece) and 35.71 PP pieces (20 years/0.56 years each PP piece). With this in mind, and Table 2 that shows the inventories for the steel turning and plastic injection processes, it is possible to calculate the amounts of steel billet, PP granulate, and electricity necessary for each process (Figure 1). By this way, the reference flows refer to the functional unit of this study.

**Table 2.** Steel turning and plastic injection inventories (1 kg).

| Input | | | |
|---|---|---|---|
| Material | Unit | Steel turning | Plastic injection |
| Electricity | MJ | 3.31 | 6.64 |
| Steel billet | Kg | 1.36 | |
| Plastic granulate | Kg | | 1.02 |
| Output * | | | |
| Material | Unit | Steel turning | Plastic injection |
| Steel piece | kg | 1 | |
| Plastic piece | kg | | 1 |

* Steel and plastic scraps leave the systems as waste: 0.36 kg steel scrap, 0.02 kg plastic scrap.

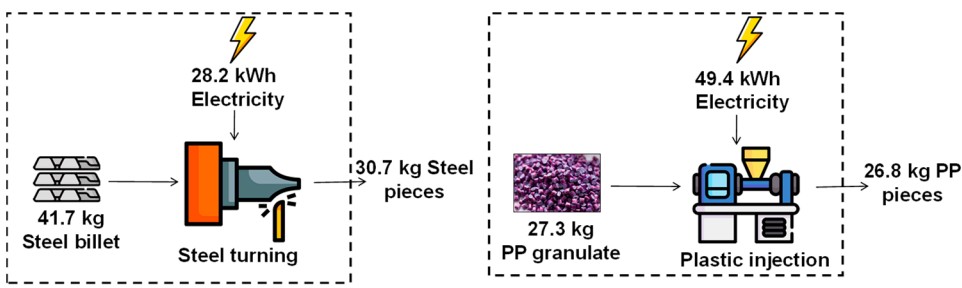

**Figure 1.** Steel turning and PP injection fundamental flows.

Steel pieces

Electricity

$$\frac{3.31 \text{ MJ electricity}}{\text{kg steel piece}} \times 13.33 \text{ steel pieces} \times \frac{2.3 \text{ kg steel pieces}}{\text{steel piece}} = 101.48 \text{ MJ} \times \frac{1 \text{ kWh}}{3.6 \text{ MJ}}$$
$$= 28.2 \text{ kWh Electricity}$$

Steel billet

$$\frac{1.36 \text{ kg steel billet}}{\text{kg steel piece}} \times 13.33 \text{ steel pieces} \times \frac{2.3 \text{ kg steel pieces}}{\text{steel piece}} = 41.7 \text{ kg steel billet}$$

PP pieces

Electricity

$$\frac{6.64 \text{ MJ electricity}}{\text{kg PP piece}} \times 35.71 \text{ PP pieces} \times \frac{0.75 \text{ kg PP pieces}}{\text{PP piece}} = 177.83 \text{ MJ} \times \frac{1 \text{ kWh}}{3.6 \text{ MJ}}$$
$$= 49.4 \text{ kWh Electricity}$$

PP granulate

$$\frac{1.02 \text{ kg PP granulate}}{\text{kg PP piece}} \times 35.71 \text{ PP pieces} \times \frac{0.75 \text{ kg PP pieces}}{\text{PP piece}} = 27.3 \text{ kg PP granulate}$$

Steel turning and plastic injection processes have fundamental flows as inputs and outputs (flows that come or go from/to the technosphere and should be provided/fed by/to other processes). If the fundamental flow is an input, it has to be connected to another process (steel billet, PP granulate, and electricity production). If the fundamental flow is an output (steel and plastic pieces), it has to leave the system or be used as an input by other processes. Fundamental flows are shown in the tables with a bold letter in this work and elementary flows (flows that come from the ecosphere (nature) to the technosphere (inputs) or flows that go from the technosphere to the ecosphere (outputs)) are in a regular letter. In Figure 1, the fundamental flows are inside the technosphere (dot lines) while the elementary flows (not shown) go from the ecosphere to the technosphere and vice versa. The LCA inventory step includes the elementary input and output flows' compilation related to the fundamental flows, which in turn refer to the functional unit. After the steel billet, electricity, and PP granulate fundamental flows' calculation, it is possible to compile the elementary inputs and outputs for both systems. Table 3 shows a simplified version of the inventories extracted from the Educational GaBi version, where only the most important elementary flows are selected. These are considered cradle-to-gate processes; therefore, they simulate the whole process from nature to the technosphere. For the steel billet production: mining and steel production; for the PP granulate production: crude oil extraction, cracking, and polymer manufacture; and for the electricity generation in Spain: the construction of the energy plants and the generation of electricity from several energy sources. In Table 3, it is possible to see that these processes have only elementary inputs and outputs, and one unique fundamental output flow (steel billet, PP granulate, and electricity).

**Table 3.** Steel billet (1 kg), PP granulate (1 kg), electricity generation from Spain grid mix (1 MJ), steam from natural gas (1 MJ), and NaOH (1 kg) inventories.

| Inputs | | | | | | |
| --- | --- | --- | --- | --- | --- | --- |
| Material | Unit | Steel billet | PP granulate | Electricity | Steam | NaOH |
| Coal | MJ | 16.37 | 2.1 | 0.42 | $2.85 \times 10^{-3}$ | 3.28 |
| Crude oil | MJ | 1.44 | 38.3 | 0.20 | $6.88 \times 10^{-3}$ | 1.24 |
| Iron | kg | 0.89 | | | | |
| Natural gas | MJ | | 26.2 | 0.54 | 1.17 | 5.32 |
| Water | kg | 323 | | | | |

| Outputs | | | | | | |
| --- | --- | --- | --- | --- | --- | --- |
| Material | Unit | Steel billet | PP granulate | Electricity | Steam | NaOH |
| Steel billet | kg | 1 | | | | |
| Polypropylene granulate | kg | | 1 | | | |
| Electricity | MJ | | | 1 | | |
| Steam | MJ | | | | 1 | |
| Sodium hydroxide | kg | | | | | 1 |
| Carbon dioxide | kg | 1.98 | 1.53 | 0.09 | 0.069 | 1.12 |
| Carbon monoxide | kg | 0.017 | | | | |
| Chloride (aq) | kg | $9.0 \times 10^{-3}$ | 0.11 | $6.67 \times 10^{-4}$ | $3.06 \times 10^{-5}$ | 0.025 |
| Methane | kg | $1.2 \times 10^{-3}$ | $6.9 \times 10^{-3}$ | $2.23 \times 10^{-4}$ | $2.23 \times 10^{-4}$ | $1.52 \times 10^{-3}$ |
| Nitrogen oxides | kg | $3.1 \times 10^{-3}$ | | $5.0 \times 10^{-4}$ | $5.18 \times 10^{-5}$ | $2.05 \times 10^{-3}$ |
| Sulphur dioxide | kg | $2.6 \times 10^{-3}$ | | $4.0 \times 10^{-4}$ | $1.85 \times 10^{-5}$ | $7.43 \times 10^{-4}$ |

Next, and as an example of the inventory calculation, the total amount of one elementary flow as crude oil is calculated.

Crude oil steel pieces

Crude oil steel billet

$$\frac{1.44 \text{ MJ Crude Oil}}{\text{kg Steel Billet}} \times 41.7 \text{ kg Steel Billet} = 60.1 \text{ MJ Crude Oil}$$

Crude oil electricity

$$\frac{0.20 \text{ MJ Crude Oil}}{1 \text{ MJ electricity}} \times \frac{3.6 \text{ MJ}}{1 \text{ kWh}} \times 28.2 \text{ kWh electricity} = 20.3 \text{ MJ Crude Oil}$$

So, the amount of crude oil elementary flow input for the steel pieces is 60.1 + 20.3 = 80.4 MJ.

And the amount of crude oil for the PP pieces will be:

Crude oil PP pieces

Crude oil PP granulate

$$\frac{38.3 \text{ MJ Crude Oil}}{\text{kg PP granulate}} \times 27.3 \text{ kg PP granulate} = 1025.9 \text{ MJ Crude Oil}$$

Crude oil electricity

$$\frac{0.20 \text{ MJ Crude Oil}}{1 \text{ MJ electricity}} \times \frac{3.6 \text{ MJ}}{1 \text{ kWh}} \times 47.4 \text{ kWh electricity} = 35.6 \text{ MJ Crude Oil}$$

So, the amount of crude oil elementary flow input for the PP pieces is 1025.9 + 35.6 = 1061.5 MJ. These calculations have been performed for all the inputs and outputs and compiled in Figure 2 for the steel and PP pieces. These data and calculations are also presented in Tables S1 and S2 on the Excel sheet "LCI & LCIA & Results" of the Excel file "Steel & PP Example" provided as Supplementary Materials to this article.

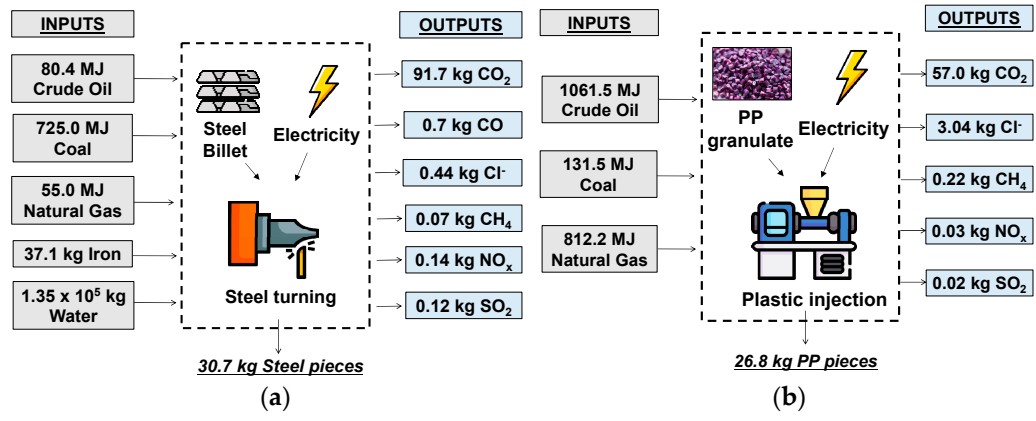

**Figure 2.** (**a**) Total inventory for steel pieces of the LCA educational example; (**b**) total inventory for PP pieces of the LCA educational example.

3.1.2. Life Cycle Impact Assessment (LCIA)

ISO 14040 [29] described this third LCA step as "the impact assessment phase of LCA is aimed at evaluating the significance of potential environmental impacts using the LCI results. In general, this process involves associating inventory data with specific environmental impact categories and category indicators, thereby attempting to understand these impacts". As has been said in the introduction of this work, LCA software (GaBi® Educational version v2022.2) automatically transforms inventory data into EIIs, and by this way, makes the LCA easier and simpler. However, for future professionals in sustainability, it is very important to understand the mechanism of this transformation, which entails a

deep knowledge of the LCA. The main process of the LCIA is the characterization step, in which the inputs and outputs of the elementary flows are transformed into EIIs. In this work, as has been said in the introduction, 16 EIIs recommended by the European Union are going to be used. The last update of the recommended EIIs made by the LCA European Platform is compiled on the Environmental Footprint v3.1. where the methodology for each EII calculation and the characterization factors used are described.

Table 4 presents the characterization factors extracted from the Environmental Footprint v3.1 methodology and used in this work. Only 11 of the 16 EIIs in Table 1 are going to be used because the inputs and outputs of this very simplified LCA only have an impact on these EIIs. Mainly, these characterization factors are the numbers that multiply the elementary flow to transform it to each EII. Characterization factors quantify the impact of the elementary flows into the different EIIs. For example, in this section, the value of the Global Warming Potential (GWP) in kg of $CO_2$ Eq. for the steel piece and for the PP piece is going to be calculated. In Table 4, it can be seen that only three elementary outputs have an impact on the GWP EII: carbon dioxide, carbon monoxide, and methane. So, the value of the GWP will be obtained after the multiplication of the value of these fluxes (Figure 2) by their respective characterization factors. The final values of the 11 EIIs for the steel and PP pieces are presented in Tables S3 and S4 on the Excel sheet "LCI & LCIA & Results" of the Excel file "Steel & PP Example" provided as Supplementary Materials to this article.

**Table 4.** Characterization factors (Environmental Footprint v3.1).

| | GWP | RI | POF | AC | EUT | EUM | HTNC | ECFW | RDM | WU | RU |
|---|---|---|---|---|---|---|---|---|---|---|---|
| **Inputs** | | | | | | | | | | | |
| Coal | | | | | | | | | | | 1 |
| Crude oil | | | | | | | | | | | 1 |
| Iron | | | | | | | | | $5.24 \times 10^{-8}$ | | |
| Natural gas | | | | | | | | | | | 1 |
| Water | | | | | | | | | | $4.30 \times 10^{-2}$ | |
| **Outputs** | | | | | | | | | | | |
| Carbon dioxide | 1 | | | | | | | | | | |
| Carbon monoxide | 1.57 | | $4.56 \times 10^{-2}$ | | | | $1.08 \times 10^{-6}$ | $2.28 \times 10^{-2}$ | | | |
| Chloride | | | | | | | $4.68 \times 10^{-8}$ | 301 | | | |
| Methane | 36.8 | | $1.01 \times 10^{-2}$ | | | | $4.85 \times 10^{-8}$ | 0.32 | | | |
| Nitrogen oxides | | $1.60 \times 10^{-6}$ | 1 | 0.74 | 4.26 | 0.39 | | | | | |
| Sulphur dioxide | | $8.00 \times 10^{-6}$ | $8.11 \times 10^{-2}$ | 1.31 | | | | | | | |

Global Warming Potential steel pieces

Global Warming Potential $CO_2$

$$91.74 \text{ kg } CO_2 \times \frac{1 \text{ kg } CO_2 \text{ Eq.}}{1 \text{ kg } CO_2} = 91.74 \text{ kg } CO_2 \text{ Eq.}$$

Global Warming Potential CO

$$0.696 \text{ kg CO} \times \frac{1.57 \text{ kg } CO_2 \text{ Eq.}}{1 \text{ kg CO}} = 1.1 \text{ kg } CO_2 \text{ Eq.}$$

Global Warming Potential $CH_4$

$$0.074 \text{ kg } CH_4 \times \frac{36.8 \text{ kg } CO_2 \text{ Eq.}}{1 \text{ kg } CH_4} = 2.72 \text{ kg } CO_2 \text{ Eq.}$$

$$\text{GWP}_{\text{Steel pieces}} = 91.74 + 1.1 + 2.72 = 95.56 \text{ kg } CO_2 \text{ Eq.}$$

Global Warming Potential PP pieces

Global Warming Potential $CO_2$

$$57.04 \text{ kg CO}_2 \times \frac{1 \text{ kg CO}_2 \text{ Eq.}}{1 \text{ kg CO}_2} = 57.04 \text{ kg CO}_2 \text{ Eq.}$$

Global Warming Potential $CH_4$

$$0.224 \text{ kg CH}_4 \times \frac{36.8 \text{ kg CO}_2 \text{ Eq.}}{1 \text{ kg CH}_4} = 8.24 \text{ kg CO}_2 \text{ Eq.}$$

$$\text{GWP}_{\text{PP pieces}} = 57.04 + 8.24 = 65.3 \text{kg CO}_2 \text{ Eq.}$$

Interpretation and Results

Figure 3 shows the contribution of the steel and PP pieces to the total 11 EII values. This way of presenting the results is necessary because the units of each EII are different (e.g., kg $CO_2$ Eq. for GWP and kg CGC-11 Eq. for ODP) and in very different orders of magnitude (Tables S3 and S4 on Supplementary Materials). From Figure 3, it is possible to conclude that the steel pieces have the highest impacts for 9 of the 11 EIIs. From Figures S1 and S2, in the document "Supplementary Materials" provided as Supplementary Materials to this article, where the contribution of each process to the total value of the steel and PP pieces is presented, it is possible to conclude that steel billet manufacture is the process with the highest impact. Regarding the PP pieces, the electricity used for the pieces' injection is the maximum contributor to five of the nine EIIs on which this system has an impact, and the PP granulate for four of the nine EIIs, although this process is the cause behind why the PP pieces have more of an impact than the steel pieces for the Ecotoxicity Freshwater (ECFW) and Fossil Resources Use (RU) EIIs.

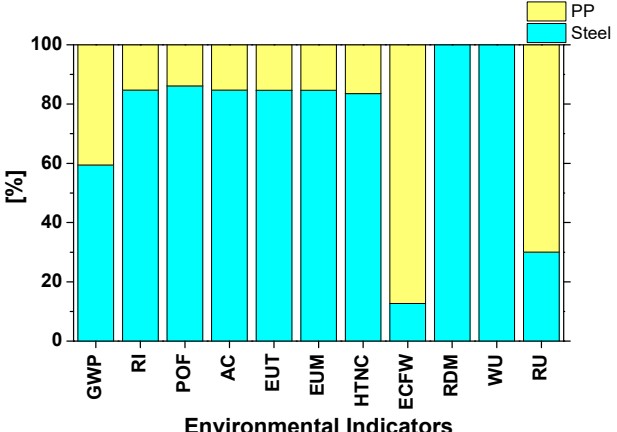

**Figure 3.** Contribution of steel and PP pieces to the total value of the 11 LCA EIIs.

3.1.3. PP End-of-Life Example

This section presents a comparative study of three different ways of treating 1 kg of a polypropylene waste piece: landfill, incineration, and mechanical recycling. This example increases the complexity of the calculations and serves as an introduction to recycling systems in LCA.

Scope and Objective of The LCA

This LCA aims to determine the environmental impacts of the three different ends of life of one PP piece. The functional unit of this study is defined as "1 kg PP waste piece end of life treatment". Like in the PP and steel pieces' example, the GaBi Educational version database is used. For the mechanical recycling process, the data were extracted from the

literature [32]. The EIIs used are the same as for the last example. In Figure 4, the three PP waste ends of life that are going to be studied are presented schematically.

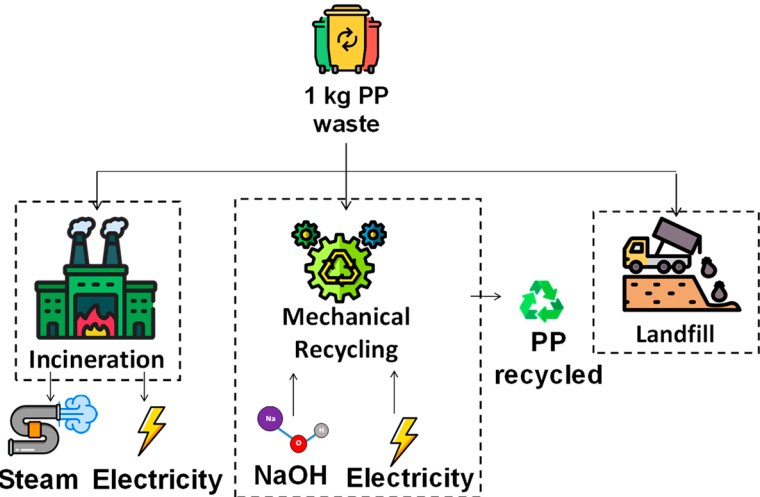

**Figure 4.** PP waste end-of-life (incineration, mechanical recycling, and landfill) LCA example.

Life Cycle Inventory (LCI)

Table 5 presents the inventories for the three end-of-life treatments considered. As can be seen, the landfill process does not require any fundamental flow or process that provides them, and only takes into account the elementary flows. The incineration PP process does not require any fundamental flows, but it releases two waste flows that can be used for other processes as fundamental flows (steam and electricity). The mechanical recycling process needs two fundamental flows (electricity and NaOH) and also produces a recycled PP granulate that can be used by other systems as the fundamental flow. LCA theory specifies that these waste and fundamental flows can be recycled in open or closed loops [12] with the aim of taking into account these flows avoid virgin materials' utilization. A closed-loop recycling process reinserts the flow into the system, and an open-loop recycling system considers that one different system outside of the limits of LCA uses waste flow as the input. In this case, the impacts should be subtracted because the systems outside of the LCA limits avoid using products from virgin materials. This is the case of this study where the steam, electricity, and PP granulate that are recycled are going to be considered as the open loop and be used by another system. Therefore, their production from virgin materials is subtracted from the initial system. So, for the inventory of the incineration of 1 kg of PP waste, it Is necessary to subtract the elementary flows for the production of 11.9 MJ of steam and 6.68 MJ of electricity (Table 5).

As an example, the inventory of crude oil for the incineration of 1 kg of PP is going to be calculated as follows:

Crude oil inventory incineration

Crude oil used in incineration

$$\frac{0.102 \text{ kg crude oil}}{\text{kg PP waste}}$$

Crude oil avoided in electricity

$$\frac{6.68 \text{ MJ electricity}}{\text{kg waste PP}} \times \frac{0.20 \text{ MJ crude oil}}{1 \text{ MJ electricity}} = \frac{1.336 \text{ MJ crude oil}}{\text{kg PP waste}}$$

Crude oil avoided in steam

$$\frac{11.9 \text{ MJ steam}}{\text{kg waste PP}} \times \frac{0.00688 \text{ MJ crude oil}}{1 \text{ MJ steam}} = \frac{0.082 \text{ kg crude oil}}{\text{kg PP waste}}$$

$$\frac{\text{Total crude oil incineration}}{\text{kg waste PP}} = 0.102 - 1.336 - 0.082 = \frac{-1.32 \text{ MJ crude oil}}{\text{kg waste PP}}$$

Thus, in the incineration of 1 kg of waste PP, 1.32 MJ of crude oil is avoided mainly due to the recycled electricity produced. For the inventory of 1 kg of PP waste mechanical recycling, it is necessary to use the inventory for the production of sodium hydroxide, electricity, and PP granulate from the virgin materials (this last part should be subtracted). The crude oil inventory of 1 kg PP waste is calculated as follows:

Crude oil inventory mechanical recycling

Crude oil used in NaOH production

$$\frac{0.017 \text{ kg NaOH}}{\text{kg waste PP}} \times \frac{1.24 \text{ MJ crude oil}}{\text{kg NaOH}} = \frac{0.021 \text{ MJ crude oil}}{\text{kg waste PP}}$$

Crude oil used in electricity generation

$$\frac{1.15 \text{ MJ electricity}}{\text{kg waste PP}} \times \frac{0.20 \text{ MJ crude oil}}{1 \text{ MJ electricity}} = \frac{0.21 \text{ MJ crude oil}}{\text{kg waste PP}}$$

Crude oil use avoided in PP granulate

$$\frac{0.69 \text{ kg PP granulate}}{\text{kg wate PP}} \times \frac{38.3 \text{ MJ crude oil}}{\text{kg PP granulate}} = \frac{26.43 \text{ MJ crude oil}}{\text{kg waste PP}}$$

$$\frac{\text{Total crude oil incineration}}{\text{kg waste PP}} = 0.021 + 0.21 - 26.43 = \frac{-26.20 \text{MJ crude oil}}{\text{kg waste PP}}$$

Thus, 1 kg of waste PP mechanical recycling avoids the use of 26.2 MJ of crude oil due to the recycled PP granulate. In Figure 5, the inventories for the incineration and mechanical recycling systems are compiled. The landfill system is not shown because it is the same as that presented in Table 5. In Tables S5 and S6, on the Excel sheet "LCI & LCIA & Results" of the Excel file "PP End of Life Example" provided as Supplementary Materials to this article, all these operations are presented. The negative values in the inputs mean that the use of the elementary flow inputs is avoided, and in the outputs, it means that the emissions of the elementary flow outputs are avoided.

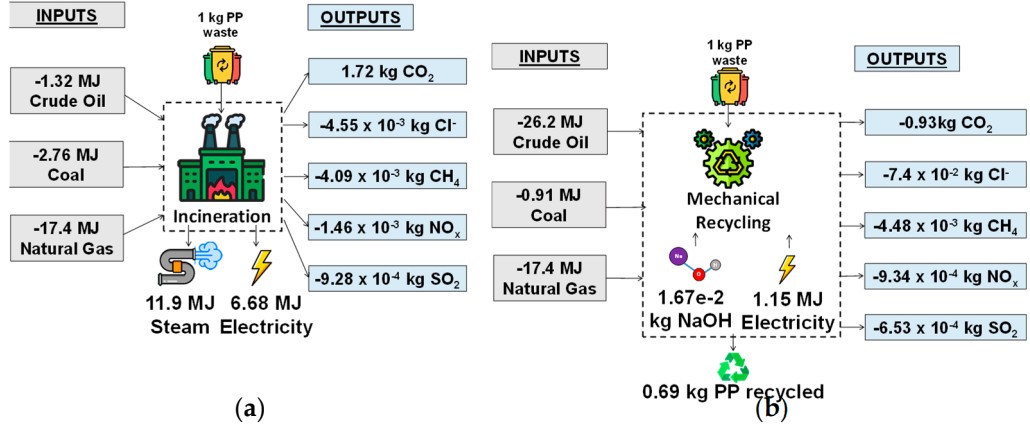

**Figure 5.** (**a**) LCI for the incineration of 1 kg of PP waste; (**b**) LCI for the mechanical recycling of 1 kg of PP waste.

**Table 5.** Inventories for PP incineration, mechanical recycling, and landfill (1 kg).

| | | Inputs | | |
|---|---|---|---|---|
| Material | Unit | PP Incineration | Mechanical Recycling | Plastic Landfill |
| Electricity | MJ | | 1.15 | |
| PP waste | kg | 1.00 | 1.00 | 1.00 |
| Sodium hydroxide | kg | | $1.67 \times 10^{-2}$ | |
| Coal | MJ | $6.76 \times 10^{-2}$ | | $6.38 \times 10^{-2}$ |
| Crude oil | MJ | 0.10 | | 0.31 |
| Natural gas | MJ | 0.15 | | 0.58 |
| | | Outputs | | |
| Material | Unit | PP Incineration | Mechanical Recycling | Plastic Landfill |
| Electricity | MJ | 6.68 | | |
| PP (recycled) | kg | | 0.69 | |
| Steam | MJ | 11.9 | | |
| Carbon dioxide | kg | 3.13 | | $5.99 \times 10^{-2}$ |
| Chloride | kg | $3.38 \times 10^{-4}$ | | $9.12 \times 10^{-4}$ |
| Methane | kg | $4.70 \times 10^{-5}$ | | $3.27 \times 10^{-4}$ |
| Nitrogen oxides | kg | $1.51 \times 10^{-4}$ | | $1.19 \times 10^{-4}$ |
| Sulphur dioxide | kg | $1.46 \times 10^{-5}$ | | $9.63 \times 10^{-5}$ |

Life Cycle Impact Assessment (LCIA) and Results

Conversion of LCI into EIIs has been performed as before, multiplying the inputs and outputs by the corresponding characterization factors. In this example, only nine EIIs are going to be used because they are the ones on which the elementary flows have an impact. The characterization factors are the same as before (Table 5). In Tables S7–S9, on the Excel sheet "LCI & LCIA & Results" of the Excel file "PP End of Life Example" provided as Supplementary Materials to this article, the results for the three different ends-of-life processes for the nine EIIs with their calculations are shown. In Figure 6, the contribution of each end of life to the EIIs' total values is shown. It is possible to conclude from this Figure 6 that mechanical recycling is the best end of life for the PP waste. Incineration also reduces the EIIs but to a lesser extent than mechanical recycling; however, it is worth noting that incineration increments the GWP value. Landfilling increments the value of the nine EIIs studied, but on the other hand, its GWP is lower than incineration. In Figures S3 and S4, in the document "Supplementary Materials" provided as Supplementary Materials to this article, the contribution of each process to the total value of the EIIs for the incineration and mechanical recycling processes is presented. From these figures, it is possible to conclude that the PP granule recycling process is the one that reduces all the EII values for the mechanical recycling process, and regarding incineration, it is the steam recycling process that reduces the most EIIs. The incineration process is the main one responsible for the high value of the GWP for the incineration end of life.

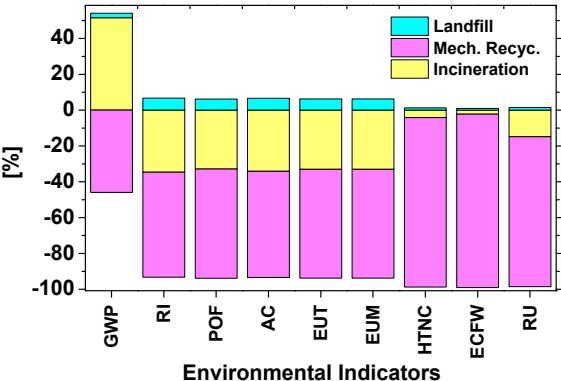

**Figure 6.** Contribution of incineration, mechanical recycling, and landfill ends of life to the total value of the nine LCA EIIs.

### 3.2. Midterm and Final Students' Marks

As explained in the method section, in order to assess the learning of the students, a midterm exam was programmed after the theoretical part of the course was finished. A total of 50% of the final course mark will comprise the mark of this exam and the other 50% will be the mark of the project performed in the practical part. Figure 7 shows the marks obtained during the last three academic courses, within the midterm and final exams.

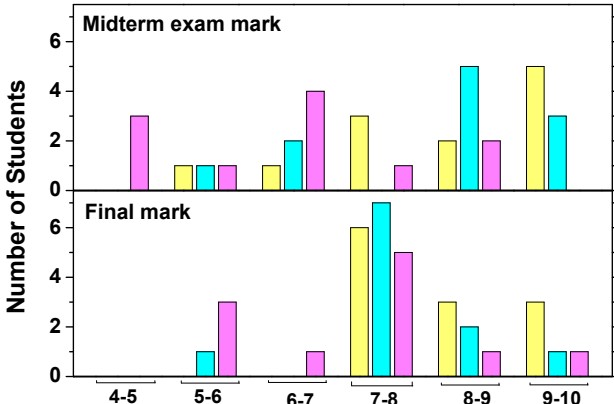

**Figure 7.** Midterm exam and final marks "Ecodesign" course for 20/21 (yellow), 21/22 (blue), and 22/23 (magenta) academic years.

As can be observed, the results obtained for the midterm exams have been highly satisfactory in the 20/21 and 21/22 academic years and only satisfactory in the 22/23 year. When the midterm exam marks were high, the final marks were also high (20/21 and 21/22 academic years), and the authors of this work believe that these final marks are good not only because the midterm exam mark contributes 50% to the final mark, but the students have assimilated the LCA fundamentals during the theoretical course part and applied them correctly into their ecodesign final projects. To achieve approval, the students had to be able to compile the elementary flows related to the fundamental flows and the functional unit, to convert the elementary flows into Environmental Impact Indicators (EIIs), and to show and interpret the LCA results.

### 4. Discussion, Conclusions, and Future Work

Sustainability is really close to the life cycle of products and processes. By knowing how to optimally evaluate (and calculate) the life cycle of something, engineers will avoid replacing parts or modifying processes that suppose a high cost in materials and natural resources in general [11,12]. In this article, two designed activities of LCA have been designed and carried out in one engineering degree to evaluate if these teaching activities favor the knowledge of the students about the LCA. Once these activities have been designed and validated over three consecutive courses, it can be concluded that these activities have favored learning about what Life Cycle Assessment is and the calculations regarding it. This can avoid incorrect interpretations of LCA results and will help future engineering processionals to choose the right options to achieve sustainable solutions.

The authors consider this paper also provides LCA teachers with didactic tools that serve to explain the theoretical foundations of LCA that the use of software can hide or hinder. To achieve the objective, two activities were developed. In the first one, the necessary steps to relate the inventory with the fundamental flows and the functional unit have been explained, and it is possible to conclude that how to adequate the inventory data to the functional unit of the LCA and how to transform the inventory data compiled on the LCI to EIIs by characterization factors by LCIA has been explained. In the second activity, which was a more complex example, the flows have been recycled in open-loop systems. As a conclusion, this exercise increases the complexity of the LCI and LCIA calculations

due to the open-loop recycling of the steam and electricity generated on incineration and the recycled PP granulate generated in the mechanical recycling process being included.

These activities provide students and LCA teachers with appropriate tools for a better understanding of LCA and interpretation of the results [20,21]. With these results, it can be concluded that spreadsheets have been a good teaching tool in order to explain the most fundamental theory of LCA, and if they have been correctly assimilated, provide the students with good tools to make an LCA and ecodesign. And on the other hand, the students have stated that to understand what they do, it is necessary to know the steps they take and their reasons [33]; therefore, this way of approaching the teaching of Life Cycle Assessment would help in that sense. The next step to this work is to design a more complex activity than those presented in this work, in which the use of three vehicles will be compared using LCA: diesel, electric, and hydrogen. Furthermore, to validate the results, samples of higher education students will be taken into account in future courses, for example, by extending this analysis for more courses.

In relation to the data obtained from the marks in the midterm and final exams, several conclusions have been reached. On the one hand, compared with the results of students in previous courses, it can be said that the grades are undoubtedly better. Throughout these last three academic years, in which the designed sequence has been implemented, all students have achieved marks greater than 5 points, both in the midterm and final test. It means that all of them have passed the course. But especially when students must apply what they had learned, throughout the development of the final project, it can be observed how the marks are higher (final marks). Understanding this final project as an evaluation test of their generalized learning, that is, the highest step of learning, it can be concluded that the sequence of activities designed and implemented for the teaching–learning of LCA is an effective didactic resource for the development of student learning in the context described in this work.

**Supplementary Materials:** The following supporting information can be downloaded at: https://www.mdpi.com/article/10.3390/su16041614/s1. **Word document.** Figure S1—Contribution of steel billet and electricity to the total value of steel pieces LCA 11 EIIs. Figure S2—Contribution of PP granulate and electricity to the total value of PP pieces LCA 11 EIIs. Figure S3—Contribution of incineration, and electricity and steam recycling to the total value of 1 kg waste PP incineration end of life LCA 11 EIIs. Figure S4—Contribution of electricity, sodium hydroxide and PP recycling to the total value of 1 kg waste PP mechanical recycling end of life LCA 9 EIIs. **Steel & PP Example_SM Excel document**. Table S1—Inventory Steel pieces example. Table S2—Inventory PP pieces example. Table S3—EIIs Steel pieces example. Table S4—EIIs PP pieces example. **PP end of life Example_SM Excel document**. Table S5—LCI Incineration 1 kg PP waste. Table S6—LCI Mechanical Recycling 1 kg PP waste. Table S7—EIIs for the incineration of 1 kg of waste PP. Table S8—EIIs for the mechanical recycling of 1 kg of waste PP. Table S9—EIIs for the landfill of 1 kg of waste PP.

**Author Contributions:** Conceptualization, A.N. and L.M.G.; methodology, A.N.; software, A.N. and I.E.; validation, A.N. and L.M.G.; formal analysis, I.E.; investigation, J.P. and E.C.; writing—original draft preparation, A.N., J.P. and E.C.; writing—review and editing, A.N., L.M.G., J.P. and E.C.; funding acquisition, A.N. and E.C. All authors have read and agreed to the published version of the manuscript.

**Funding:** This research received no external funding.

**Institutional Review Board Statement:** Not applicable.

**Informed Consent Statement:** Not applicable.

**Data Availability Statement:** Data are contained within the article and Supplementary Materials.

**Acknowledgments:** Authors gratefully acknowledge for Erasmus+ KA2 Knowledge Alliances for Higher Education PackAll N° 612212-EPP-1-2019-1-ES-EPPKA2-KA. We also thank to BEAGLE research group of Gobierno de Aragón, and also to IUCA institute of research.

**Conflicts of Interest:** The authors declare no conflicts of interest.

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
