# Peer review of "Life Cycle Assessment in Higher Education: Design and Implementation of a Teaching Sequence Activity"

_sustainability, doi:10.3390/su16041614_

Round 1
Reviewer 1 Report
Comments and Suggestions for Authors
This paper is to explain how to enhance students’ knowledge of chemical and material engineering in a learning courses, for understanding the concept of Life Cycle Assessment (LCA) in the sustainability.
In the introduction, this paper has clearly highlighted what the problem they concerned. Such as the Life Cycle Assessment is “A teaching aiming to make the same calculations as software but in a very simplified way” (line 61-62).
The section two is the Materials and Methods, which has showed the sample with 34 students, and teaching procedure from generalization of learning to 60 hours of on-site lessons, and develop a project from students’ learning concept.
The section three has showed the fruitful Results, which all according to the ISO 14040 regulation. And the section four the Discussion, conclusions and future work has also described.
In my opinion, this paper is interesting, the contents have shown the detailed teaching process, how to approach the goal of course. This paper is fully related to a teaching practical meaning. Therefore, I have some suggestions for the authors.
The methodology is to show how to collect data and analyze data. But this paper is lack in this part.
The results are to show what the significant research result has revealed, not a detailed mention to readers, when I read that was so boring, too technique not friendly with general readers.
And discussion has no theory development, which is making this paper far away an academic field.
The concussion is to say this study has approach the goal of the study, and the future work is to say what the direction other researches can do, but this paper is missing.
And especially, I do not know this paper how to approach the goal of sustainability. In the end of paper, I still can not find any more.
According to my suggestion, this paper is difficult to fit the goal of journal. The authors have encouraged to do more efforts for revision. Keep going.
Author Response
The authors would like to thank the reviewers for their comments, which have undoubtedly helped to improve the work. Below we respond in bold to each of the comments, in the best way possible.
Reviewer 1:
This paper is to explain how to enhance students’ knowledge of chemical and material engineering in a learning courses, for understanding the concept of Life Cycle Assessment (LCA) in the sustainability. In the introduction, this paper has clearly highlighted what the problem they concerned. Such as the Life Cycle Assessment is “A teaching aiming to make the same calculations as software but in a very simplified way” (line 61-62). The section two is the Materials and Methods, which has showed the sample with 34 students, and teaching procedure from generalization of learning to 60 hours of on-site lessons, and develop a project from students’ learning concept. The section three has showed the fruitful Results, which all according to the ISO 14040 regulation. And the section four the Discussion, conclusions and future work has also described.
In my opinion, this paper is interesting, the contents have shown the detailed teaching process, how to approach the goal of course. This paper is fully related to a teaching practical meaning. Therefore, I have some suggestions for the authors.
The authors thank all these kind comments.
The methodology is to show how to collect data and analyze data. But this paper is lack in this part.
We appreciate this comment because certainly, in the methodology we have focused on explaining how we design the teaching sequence and how we carry it out. To complete the data collection, we have added a paragraph detailing the evidence collected.
The results are to show what the significant research result has revealed, not a detailed mention to readers, when I read that was so boring, too technique not friendly with general readers.
We are really sorry that Reviewer 1 found this part of the article boring. However, we want to emphasize that the design of the teaching sequence is a result in itself. In fact, it is the most important part of the work, since it will allow readers to learn about a teaching methodology about LCA that has proven to be valid for students' knowledge of the topic. Also, one of the main objectives of this article is to provide teachers in the field of LCA with the necessary tools so that their students obtain a deep knowledge of this environmental tool. The detailed description of these tools may be boring for an average reader, but very useful for the reader to whom the work is really directed: teachers in the field of LCA. That is why we consider that despite the reading may be difficult for an average reader, if we eliminated parts of the description, the work would no longer meet its objectives.
And discussion has no theory development, which is making this paper far away an academic field.
We appreciate this comment. Reviewer 1 is right. However, as it has been said in the manuscript and before, there are two main objectives in this work, based on the necessity described, two examples of the application of LCA have been designed in a very detailed way. On the other hand, to analyze the impact of the application of these activities sequence into the student´s marks, along three years of teaching. So, regarding this second objective, there has been a research about how the LCA examples development have influenced into the students learning. However, we assume that the main part of the manuscript is devoted to the two LCa examples explaining and the research part is by this part. To amend this, we have supported the conclusions with research that reinforces and validates our results.
The concussion is to say this study has approach the goal of the study, and the future work is to say what the direction other researches can do, but this paper is missing. And especially, I do not know this paper how to approach the goal of sustainability. In the end of paper, I still can not find any more.
In the field of education, sustainability can be approached from several points of view. The work we present addresses sustainability first and foremost from the proven importance of quality education in higher education, in order to achieve SDG number 4. But above all, from the field of study of this work, chemical engineering, sustainability is closely related to the life cycle of products and processes. Knowing how to optimally evaluate (and calculate) the life cycle of something will avoid replacing parts or modifying processes, which would entail a high cost in materials with what this means for the environment and natural resources in general.
As already stated in the introduction to this article, the authors of this work have taught Ecodesign and LCA for several years. LCA was explained during those years with computer software, since these are the ones used in the industry to perform LCA. However, throughout these years, the authors have observed that the use of the software, despite facilitating the performance of LCA, impeded theoretical knowledge of the tool, which leads to simplification and misinterpretation of the results. This mistake could lead to future professional to choose option that seem good options to reach sustainability but which, in reality, lead to choosing the wrong options to achieve sustainability and even worsen the environmental impact.
That is why it was concluded that it was necessary to focus on the theoretical aspects of the LCA before using the software. If you do not know how an LCA is performed and what calculations are behind the software, it is impossible to make a complex and accurate interpretation of the results. Therefore, it is considered that this article, which provides teachers with precise tools for teaching LCA, contributes to future professionals in the field of sustainability knowing the environmental quantification tool LCA in depth and thus being able to make correct interpretations. This work will help them to choose the right path in numerous products and processes to achieve sustainability. For this reason, in order to make it explicit, we have added a phrase in the introduction and in the conclusions so that the relationship between our work and sustainability is clearly linked.
According to my suggestion, this paper is difficult to fit the goal of journal. The authors have encouraged to do more efforts for revision. Keep going.
Thank you very much to the reviewer 1 for his/her ideas and suggestions that we have tried to answer and develop on the above points. We really believe that with this review the work has improved and it is closer to the aims of the sustainability journal.
Reviewer 2 Report
Comments and Suggestions for Authors
The paper is of interest and could be improved previous to its publication.
This is a good work and the topic is very important.
Please, revise the language, the space between words in the document and the extreme use of some words in the paper. That is the case of "but"in some parts of the introduction section.
Good luck

Comments on the Quality of English LanguageThe language must be carefully reviewed
Author Response
The authors would like to thank the reviewers for their comments, which have undoubtedly helped to improve the work. Below we respond in bold to each of the comments, in the best way possible.
Reviewer 2:
The paper is of interest and could be improved previous to its publication.
This is a good work and the topic is very important.
Please, revise the language, the space between words in the document and the extreme use of some words in the paper. That is the case of "but"in some parts of the introduction section.
Thank you so much to the Reviewer 2 for his/her words. Attending this comment, we have substitute some “but” by other expressions.
Nowadays, the study of the strong relationship between chemical engineering and sustainable development is a matter of a great relevance. Any initiative that encourages the interest of chemical engineers in sustainable development deserves a great attention. Consequently, the paper deals with a very interest topic. Despite the importance of the document, in my opinion some minor changes should be taken into account prior to its publication.
We would thank these comments.
In general : Please, use an English native speaker to review the language.
Attending this comment, an English native revisor has review the text previous this second submission.
Abstract.
This sentence is incomplete or deserves to be redacted again in a better way: "In this work, a study in which the reflected on the lack of knowledge of engineering students, in relation to the life cycle analysis of products and processes
We agree with the reviewer, and thus we have modified the sentence by “In this work, we present a study about the life cycle analysis of products and processes knowledge of engineering students”.
Avoid the use of Acronyms in the abstract. LCA appears before its significate.
We agree. We have substitute LCA by Life Cycle Assessment.
Introduction
This is a very strong statement: "Perhaps the only study that has provided 80 the basic LCA calculations is Cosme et al.[19]. Please avoid the use of such as statements.
We agree, therefore we have modified the sentences by: “One of the few studies that has provided 80 the basic LCA calculations is Cosme et al. [19].”
The objectives of the paper are not well explained. Please, revise and rewrite from line 89 to 94. Please, insert a paragraph describing how the objectives are going to be achieved more clearly.
As reviewer ask for, we have simplified the two main objectives of the study, in order to clarify the aim of this paper.
In the materials and methods, we have added an addition paragraph linked these objectives with the way to achieved them.
Materials and methods
Is the sample used representative of the total? This need to be justified.
We agree. We have added this note in the text.
Results:
Review the lines on Table l .
The formulation used in not clear. For example: The use of (Table 2) in lines 203 and 206 means that the previous division must be multiplied by the numbers on this table? The observation above includes also lines 253, 257 and the others were formulations include (Table NO) or line 318 were it is possible to see (Figure 2) inserted in a mathematical expression.
The reference in the formula to the table is where it has been extracted the data. For example, in line 203, the reference to Table 2 I because 3.31 MJ/Kg data have been extracted from this Table. This way of referencing data has been used with the aim of a better explaining. Because Reviewer 2 thinks that this way of explaining could be difficult for the reader, these references have been removed from the text. Thank you to the reviewer 2 for the suggestion that has improved the text.
Conclusions:
This section should be more consistent Could be the influence of other external factors the consequence of the achievement of higher marks comparing with previous courses.
We have fundamental some of the results with bibliography. To avoid this degree of freedom, the analysis has been carried out for three consecutive years. In this way, although the student sample is different each year, sufficient results are obtained to be able to state that the improvement in learning about LCA is due to the variation in the teaching methodology. Without a doubt, future work should validate this result by expanding the sample, for example by extending this analysis for more courses.
Please, explain in detail how this study contributes to the sustainable development in this degree.
As already stated in the introduction to this article, the authors of this work have taught Ecodesign and LCA for several years. LCA was explained during those years with computer software, since these are the ones used in the industry to perform LCA. However, throughout these years, the authors have observed that the use of the software, despite facilitating the performance of LCA, impeded theoretical knowledge of the tool, which leads to simplification and misinterpretation of the results. This mistake could lead to future professional to choose option that seem good options to reach sustainability but which, in reality, lead to choosing the wrong options to achieve sustainability and even worsen the environmental impact.
That is why it was concluded that it was necessary to focus on the theoretical aspects of the LCA before using the software. If you do not know how an LCA is performed and what calculations are behind the software, it is impossible to make a complex and accurate interpretation of the results. Therefore, it is considered that this article, which provides teachers with precise tools for teaching LCA, contributes to future professionals in the field of sustainability knowing the environmental quantification tool LCA in depth and thus being able to make correct interpretations. This work will help them to choose the right path in numerous products and processes to achieve sustainability. For this reason, in order to make it explicit, we have added a phrase in the introduction and in the conclusions so that the relationship between our work and sustainability is clearly linked.
Reviewer 3 Report
Comments and Suggestions for Authors
Based on LCA teaching practice in higher education and SDGs, the study provides an effective LCA teaching sequence with relevant teaching examples and resources. Therefore, it was equipped with important practical significance. However, there are some flaws in this paper, and I hope the authors can revise them.
Firstly, the "Introduction" is too long and a little bit illogical. The authors wrote three paragraphs to talk about the practical need for the study but only one paragraph to state the theoretical necessity of the study. This imbalanced arrangement is not good for an academic paper whose theoretical orientation should have been highlighted. More balanced proportions between the two parts should be achieved in the revised version. Please directly tell the readers what the research purposes in the beginning of this part.
Secondly, in the "2.2 Procedure", the content of the first paragraph has a weak connection with the procedure of the study. The paragraph seems to be relevant to the necessity of the study. The authors can transfer this content to the "Introduction" part or simplify it.
Fourthly, tenses were not used correctly. The simple past tense or the present perfect tense should be used when the authors want to cite previous studies.
Last, more theoretical and methodological information about the LCA is desired in the revised version.
Comments on the Quality of English LanguageBased on LCA teaching practice in higher education and SDGs, the study provides an effective LCA teaching sequence with relevant teaching examples and resources. Therefore, it was equipped with important practical significance. However, there are some flaws in this paper, and I hope the authors can revise them.
Firstly, the "Introduction" is too long and a little bit illogical. The authors wrote three paragraphs to talk about the practical need for the study but only one paragraph to state the theoretical necessity of the study. This imbalanced arrangement is not good for an academic paper whose theoretical orientation should have been highlighted. More balanced proportions between the two parts should be achieved in the revised version. Please directly tell the readers what the research purposes in the beginning of this part.
Secondly, in the "2.2 Procedure", the content of the first paragraph has a weak connection with the procedure of the study. The paragraph seems to be relevant to the necessity of the study. The authors can transfer this content to the "Introduction" part or simplify it.
Fourthly, tenses were not used correctly. The simple past tense or the present perfect tense should be used when the authors want to cite previous studies.
Last, more theoretical and methodological information about the LCA is desired in the revised version.
Author Response
The authors would like to thank the reviewers for their comments, which have undoubtedly helped to improve the work. Below we respond in bold to each of the comments, in the best way possible.
Reviewer 3:
Based on LCA teaching practice in higher education and SDGs, the study provides an effective LCA teaching sequence with relevant teaching examples and resources. Therefore, it was equipped with important practical significance. However, there are some flaws in this paper, and I hope the authors can revise them.
Thanks for the comments.
Firstly, the "Introduction" is too long and a little bit illogical. The authors wrote three paragraphs to talk about the practical need for the study but only one paragraph to state the theoretical necessity of the study. This imbalanced arrangement is not good for an academic paper whose theoretical orientation should have been highlighted. More balanced proportions between the two parts should be achieved in the revised version. Please directly tell the readers what the research purposes in the beginning of this part.
As the referee indicates, the “Introduction” section begins now with the objective of our study. Also we have shortened the text as it has been required. And also we have trying to equilibrate this paragraphs.
We especially appreciate this comment since, together with the following comment, we consider that it has helped build a much more understandable and direct introduction for the reader.
Secondly, in the "2.2 Procedure", the content of the first paragraph has a weak connection with the procedure of the study. The paragraph seems to be relevant to the necessity of the study. The authors can transfer this content to the "Introduction" part or simplify it.
As the referee indicates, the first paragraph of the “Procedure sections has been moved to the “introduction” section.
Fourthly, tenses were not used correctly. The simple past tense or the present perfect tense should be used when the authors want to cite previous studies.
The hole manuscript has been reviewed attending to the English idiom and specially to the tense of the verbs.
Last, more theoretical and methodological information about the LCA is desired in the revised version.
Reviewer 3 suggests that more theoretic and methodologic introduction to LCA is necessary, but we strongly believe that this could make article reading for standards readers more difficult and boring. For this reason, we have introduced short explanations of each step of the life cycle assessment and the beginning of each section. However, in order to improve the LCA explanation we have introduced more information into the beginning of each section. Thank you to the reviewer 3 for the suggestion.
Round 2
Reviewer 1 Report
Comments and Suggestions for Authors
This is my seconded time to read the submitted paper.
The revised version has corrected what I suggestion on first version.
I think this revised version is suitable to the journal.
Author Response
The authors would like to greatly appreciate the reviewer's comments,which have helped to improve our work considerably.
Reviewer 2 Report
Comments and Suggestions for Authors
The authors have correctly addressed the comments made in the initial review.
Author Response

(The authors gave the same response as above.)

Reviewer 3 Report
Comments and Suggestions for Authors
The authors did careful revisions. I recommended it to be published after a copyediting.
Comments on the Quality of English LanguageThe authors did careful revisions. I recommended it to be published after a copyediting.
Author Response

(The authors gave the same response as above.)
